# Learning Agent Representations for Ice Hockey

**Guiliang Liu**[1], **Oliver Schulte**[1,3], **Pascal Poupart**[2,4], **Mike Rudd**[2,4], **Mehrsan Javan**[3]

[1]School of Computing Science, Simon Fraser University
[2]Cheriton School of Computer Science, University of Waterloo
[3]SLiQ Lab, Sportlogiq
[4]Vector Institute

`gla68@sfu.ca, oschulte@cs.sfu.ca`
`{ppoupart,jmrudd}@uwaterloo.ca, mehrsan@sportlogiq.com`

## Abstract

Team sports is a new application domain for agent modeling with high real-world impact. A fundamental challenge for modeling professional players is their large number (over 1K), which includes many bench players with sparse participation in a game season. The diversity and sparsity of player observations make it difficult to extend previous agent representation models to the sports domain. This paper develops a new approach for agent representations, based on a Markov game model, that is tailored towards applications in professional ice hockey. We introduce a novel framewwork *player representation via player generation*, where a variational encoder embeds player information with latent variables. The encoder learns a context-specific shared prior to induce a *shrinkage effect* for the posterior player representations, allowing it to share statistical information across players with different participation rates. To capture the complex play dynamics in sequential sports data, we design a Variational Recurrent Ladder Agent Encoder (VaRLAE). This architecture provides a contextualized player representation with a hierarchy of latent variables that effectively prevents latent posterior collapse. We validate our player representations in three major sports analytics tasks. Our experimental results, based on a large dataset that contains over 4.5M events, show state-of-the-art performance for our VarLAE on facilitating 1) identifying the acting player, 2) estimating expected goals, and 3) predicting the final score difference.

## 1 Introduction

Team sports define complex interactions with a rich action space and a heterogeneous state space [1]. As more and larger event stream datasets for professional sports matches become available, advanced machine learning algorithms have been applied to model the complex dynamics of sports games. These models facilitate many applications with high real-world impact, such as predicting match outcomes [2], assessing the performance of players or teams [3, 4, 5, 6], and recommending tactics to coaches [7]. However, previous models often pool observations of different players without capturing the specific roles and behaviors of each athlete. Neglecting the special characteristics of individual players significantly compromises the model performance for the application tasks [8].

A promising approach to incorporating player information into sports statistics is *deep agent representation*. Learning agent representation in a sports game raises new challenges that require a novel solution for several reasons: 1) Previous methods commonly learn a representation of agents' *policy* [9, 10, 11, 12, 13, 14]. The policy embedding methods model only how a player acts in a *given* match context, and thus fail to capture the statistical information about which match contexts players are likely to act in, as well as their immediate outcomes (rewards). 2) Although policy embeddings can facilitate the construction of an artificial agent in *control* problems, the goal of agent representation

in sports analytics is *predicting* game-relevant events using player identity information [8, 7]. 3) Previous agent representation models are often validated in virtual games instead of physical sports. They assume a limited number of agents (usually less than 10) and *sufficient* observations for each of them. In contrast, professional sports leagues often have upwards of one-thousand players, and many bench (backup) players play only a few (less than 20) games in a season. The large number of agents and unbalanced observations result in unstable embedding models that overfit to players with high participation during training, which undermines both the training convergence and the predictive performance of underlying policy representations [11].

This work describes a novel player representation framework that learns *player representations via player generation.* Conditioning on the current game context, the generation model predicts the distribution of the currently acting player. During this process, we learn a distribution of latent variables as the context-specific prior, allowing a neural encoder to derive an approximate posterior as a *contextualized representation* for the observed player. We train the encoder by maximizing an Evidence Lower Bound (ELBo, Equation (9)) that moves the posterior for each player toward the prior mode. This *shrinkage effect* [15] is ideal for player representations, because: 1) It allows information to be transferred between the observations of different players and draws closer the representations of players who share statistical similarities. 2) A shrinkage estimator prevents the encoder from overfitting to players with high participation, allowing our representation to generalize to the diversity and the sparsity of context-aware player distributions.

Following our representation framework, we design a Variational Recurrent Ladder Agent Encoder (VaRLAE) to learn player representations. VaRLAE utilizes a ladder structure [16] where, in each layer, latent variables condition on a context variable and the representations from upper layers, based on a causal dependence of latent variables that follows a Markov Game Model [17]. To incorporate play history into player representations, VaRLAE applies a recurrent network to sequential sports data. The ladder hierarchy of latent variables embeds both context and player information, which effectively improves the generative performance and prevents posterior-collapse [16, 18].

We demonstrate the generative performance of our VaRLAE on a massive National Hockey League (NHL) dataset containing over 4.5M events. VaRLAE achieves a leading identification accuracy (>12% for the players with sparse participation) over other deterministic encoders and policy representation models. To study how much the learned player representations improve downstream applications, we assess two major tasks in sports analytics: Predicting expected goals and final match score differences. Empirical results show the improvement in predictive accuracy after incorporating the embeddings generated by VaRLAE as input.

## 2 Related Work

We describe the previous works that are most related to our approach.

**Agent Representation.** Previous works [19, 20, 12] represented an agent by imitating its policy in the learning-from-demonstration [21] setting. Some recent works [11, 9, 22, 23] extended behavioral modelling to a multi-agent system. Multi-agent representations include modeling interactions between agents and their opponents' behavior for reasoning about a specific goal, such as winning a poker or video game. These interactive policy representations scale poorly to a large number of agents [11, 24]. To identify the agent representations, [9] introduced a triplet loss that encourages an exponential distance between deterministic embeddings for different agents. Policy representations have been successful for the design of artificial agents in control problems, such as games or robot control. However, real human players in professional sports cannot be controlled like artificial agents. For sports analytics, the goal of representing players is *predicting* game-relevant events using player identity information [8, 7]. This goal often requires modeling as many aspects of a players' behavior as possible, so we also represent not only the agents' policies, but also the distribution of states and rewards associated with them. Some recent works [25, 26] have proposed modeling the trajectories of multiple agents to predict their future paths. They applied a Generative Adversarial Networks (GAN) to discriminate fake from real trajectories without explicitly learning agent representations, and thus their models are not directly applicable to embed player identities.

**Variational Encoders.** Variational encoders apply a set of latent variables $\mathbf{z}$ to embed the observations $\boldsymbol{o}$. The latent variables form a disentangled representation where a change in one dimension corresponds to a change in one factor of variation [27]. Such representations are often

interpretable [28, 29] and have been applied to embed information in multiple domains ( e.g. robot skills [30] and task environment [31]). The learned representations can significantly facilitate generating many kinds of complicated data, such as images [32, 33] and action trajectories [34, 35]. To learn these representations, the Variational Auto-Encoder (VAE) maximizes an Evidence Lower Bound (ELBo): $\log p(\boldsymbol{o}) \geq \mathbb{E}_{q(\mathbf{z}|\boldsymbol{o})}\Big[\log p(\boldsymbol{o}|\mathbf{z})\Big] - \mathcal{D}_{\text{KL}}(q(\mathbf{z}|\boldsymbol{o})\|p(\mathbf{z}))$. The traditional VAE design uses a Gaussian encoder and a neural decoder to model the approximate posterior $q(\mathbf{z}|\boldsymbol{o})$ and the likelihood function $p(\boldsymbol{o}|\mathbf{z})$ respectively. The Conditional VAE (CVAE) [33] is an extension that conditions the generation process on some external environment variables. To model sequential data, the Variational Recurrent Neural Network (VRNN) [36] combines CVAE with LSTM recurrence. The Ladder VAE (LVAE) models dependencies among latent variables with a top-down structure that effectively prevents posterior collapse [16].

## 3 Player Representation Framework

We introduce the contextual variables for ice hockey and the player representation framework.

### 3.1 Contextual Variables for Ice Hockey Players

We model the ice-hockey games with a Markov Game Model [17]: $G = (\mathcal{S}, \mathcal{A}, \mathcal{T}, \mathcal{R}, \Omega)$. At each time step $t$, an agent performs an action $\mathbf{a}_t \in \mathcal{A}$ at a game state $\mathbf{s}_t \in \mathcal{S}$ after receiving an observation $\boldsymbol{o}_t \in \Omega$. This process generates the reward $\boldsymbol{r}_t \sim R(\mathbf{s}_t, \mathbf{a}_t)$ and the next state $\mathbf{s}_{t+1} \sim \mathcal{T}(\mathbf{s}_t, \mathbf{a}_t)$. For each game, we consider event data of the form $[(\boldsymbol{o}_0, pl_0, \mathbf{a}_0, \boldsymbol{r}_0), (\boldsymbol{o}_1, pl_1, \mathbf{a}_1, \boldsymbol{r}_1), \ldots, (\boldsymbol{o}_t, pl_t, \mathbf{a}_t, \boldsymbol{r}_t), \ldots]$: at time $t$, after observing $\boldsymbol{o}_t$ , player $pl_t$ takes a turn (possesses the puck) and chooses an action $\mathbf{a}_t$, which produces a reward $\boldsymbol{r}_t$ (goal score) .

The play dynamics for the acting player $pl_t$ can be captured by the following contextual variables: 1) The *game state* $\mathbf{s}_t$ describes the game environment where the action is performed. To alleviate the partial observability of observations, a game state includes the game history: $\mathbf{s}_t \equiv (\boldsymbol{o}_t, \boldsymbol{r}_{t-1}, \mathbf{a}_{t-1}, pl_{t-1}, \boldsymbol{o}_{t-1}, \ldots, \boldsymbol{o}_0)$. We utilize the RNN hidden states $\mathbf{h}_{t-1}$ to capture the game history [37], so $\mathbf{s}_t \equiv (\boldsymbol{o}_t, \mathbf{h}_{t-1})$. 2) The *action* $\mathbf{a}_t$ records the action of the on-puck player. 3) The *reward* $\boldsymbol{r}_t$ denotes whether a goal is scored after performing $\mathbf{a}_t$. As in general RL, the sequence state-action-reward can be interpreted *causally* in sports: the player makes observations summarized in a state signal $\mathbf{s}_t$, which influences his action $\mathbf{a}_t$; together with the environment, the state and action influence whether the player scores a goal. The corresponding causal graph is $\mathbf{s}_t \rightarrow \mathbf{a}_t \rightarrow \boldsymbol{r}_t$.

### 3.2 Player Representation via Player Generation

We introduce our framework of learning player representations through modeling a player generation distribution: $p(pl_t|\mathbf{s}_t, \mathbf{a}_t, \boldsymbol{r}_t)$. This distribution describes the *where and what* of a player's characteristics: what game states they tend to act in, what their actions are, and what immediate outcomes they achieve. Inspired by previous work [30, 31, 38], we utilize latent variables $\mathbf{z}_t$ as a representation of game context, which can be decoded to the distribution of current acting players:

$$p(pl_t|\mathbf{s}_t, \mathbf{a}_t, \boldsymbol{r}_t) = \int p(pl_t|\mathbf{z}_t)p(\mathbf{z}_t|\mathbf{s}_t, \mathbf{a}_t, \boldsymbol{r}_t)\mathrm{d}\mathbf{z}_t \tag{1}$$

where, *before* observing the acting player $pl_t$, the **context-aware** prior $p(\mathbf{z}_t|\mathbf{s}_t, \mathbf{a}_t, \boldsymbol{r}_t)$ models which players are likely to perform $\mathbf{a}_t$ under $\mathbf{s}_t$ and receive $\boldsymbol{r}_t$. The motivation for a contextualized representation is that the behavior of sophisticated agents, like professional players, is highly sensitive to context and it is difficult to learn a fixed representation that can adequately describe a player's tendencies under every game context. *After* observing the target player $pl_t$, we learn an approximate posterior $q(\mathbf{z}_t|\mathbf{s}_t, \mathbf{a}_t, \boldsymbol{r}_t, pl_t)$ as a **contextualized player representation**, so the complete generative process becomes:

$$p(pl_t|\mathbf{s}_t, \mathbf{a}_t, \boldsymbol{r}_t) \approx \int p(pl_t|\mathbf{z}_t)q(\mathbf{z}_t|\mathbf{s}_t, \mathbf{a}_t, \boldsymbol{r}_t, pl_t)\mathrm{d}\mathbf{z}_t \tag{2}$$

During training, our ELBo objective (Equation 9) induces a **shrinkage effect** through minimizing the Kullback–Leibler (KL) Divergence between the posterior representation for each individual player $q(\mathbf{z}_t|\mathbf{s}_t, \mathbf{a}_t, \boldsymbol{r}_t, pl_t)$ and a context-specific prior $p(\mathbf{z}_t|\mathbf{s}_t, \mathbf{a}_t, \boldsymbol{r}_t)$. A shrinkage estimator is ideal for

learning player representation for team sports because 1) it has strong statistical properties that allow information to be transferred between the observations of different players, or of the same player in different game contexts. The shrinkage effect becomes stronger for players who share many statistical similarities under a game context, which draws their representations closer. This naturally formalizes our intuition that *statistically similar players are assigned similar representations under similar game contexts.* 2) The Shrinkage term also works as a regularizer that prevents the player representations from overfitting to some frequently-present players. Compare to a deterministic auto-encoder, the stochastic shrinkage estimator can generalize to more agents with sparse observations.

In contrast to a policy representation [9, 38, 10] that captures how a player acts in a *given* state, our player representation also models when and where they act, and how successful their actions tend to be. Since our context includes actions, states, and rewards, our learned embeddings reflect how players differ in all three of these dimensions.

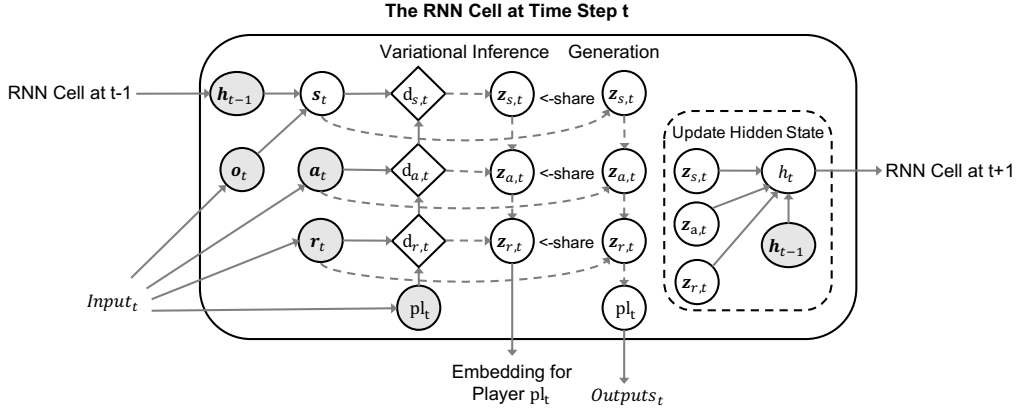

Figure 1: Our VaRLAE model includes a conditional Ladder-VAE at every RNN cell. VaR-LAE applies a top-down dependency structure ordered as the sports causal relationship (Section 3.1). The thick/dash lines denote logical-functions /stochastic-dependence. The shaded nodes are given.

# 4 A Variational Agent Encoder for Learning Agent Representations

Our VaRLAE architecture VaRLAE (Figure 1) for learning player representations follows our our representation framework (Section 3.2). We specify the generation and inference computations.

**Generation.** To incorporate match history into the game state $s_t$, the generation model combines a CVAE with recurrence following [36]. Previous works [18, 39], however, showed that a high-capacity decoder often causes the problem of *posterior collapse*, where the generator computes the distribution of the currently acting player from context variables without relying on latent variables. To make latent variables embed meaningful player information, we introduce a separate latent variable $z_{c,t}$ for each component of our context, organized in a ladder VAE [16] (Figure 1). To utilize the context information (e.g. game state $s_t$ at the top layer), the generator is forced to utilize hierarchical latent variables. (Detailed analysis appears in the appendix ). The dependency of the ladder latent variables $(z_{s,t} \rightarrow z_{a,t} \rightarrow z_{r,t})$ follows the causal dependencies $s \rightarrow a \rightarrow r$ in a Markov game (Section 3.1).

*Latent Priors.* The priors are computed as a function of the game context, rather than a context-independent Gaussian distribution. At each time step, we compute the conditional priors (of player representations) following a Gaussian distribution at each layer:

$$p(\mathbf{z}_{c,t}|c_t, \mathbf{z}_{(+),t}) = \mathcal{N}(\boldsymbol{\mu}_{c,t}^p, \boldsymbol{\sigma}_{c,t}^p) \text{ where } [\boldsymbol{\mu}_{c,t}^p, \boldsymbol{\sigma}_{c,t}^p] = \psi^{prior,c}[\psi^c(c_t, \psi^{\mathbf{z}(+)}(\mathbf{z}_{(+),t}))] \quad (3)$$

The context variable $c \in \{\mathbf{s}, \mathbf{a}, \boldsymbol{r}\}$, and $\mathbf{z}_{(+),t}$ denotes latent variables from the upper layer. We omit $\mathbf{z}_{(+),t}$ at the top layer. The neural functions $\psi^{(\cdot)}$ are implemented by a Multi-Layer Perceptron (MLP) with batch normalization at each layer. We use linear and soft-plus activation function to compute the Gaussian parameters $\mu$ and $\sigma$ respectively.

Given the latent variables $\mathbf{z}_{\boldsymbol{r},t}$ sampled from the context-specific Gaussian prior, our model generates the label of the acting (or on-puck) player as follows:

$$p(pl_t|\mathbf{z}_{\boldsymbol{r},t}) = Categorical(\theta_{1,t}, \dots, \theta_{N,t}), \text{ where } \boldsymbol{\theta}_t = \phi\{\psi^{dec}[\psi^{\mathbf{z}_{\boldsymbol{r}}}(\mathbf{z}_{\boldsymbol{r},t})]\} \tag{4}$$

The neural function $\psi^z$ extracts features from latent variable $\mathbf{z}_{\boldsymbol{r},t}$. These features are then sent to another decoder function $\psi^{dec}$. Given the decoder outputs, we apply a softmax function $\phi$ to generate categorical parameters $\boldsymbol{\theta}_t$. The output $\theta_{i,t}$ represents the probability of player $i$ acting at time $t$.

**Inference.** We apply variational inference to derive an objective function for estimating the parameters of our hierarchical model. Our VaRLAE defines a top-down dependency structure by utilizing the hierarchical priors and approximate posteriors on latent variables to derive an approximate log-likelihood function for the observed data [16, 32].

After observing player $pl_t$, we first implement a deterministic upward pass to compute the approximate likelihood contributions. Conditioning on reward variable $\boldsymbol{r}_t$, the bottom layer computes:

$$[\hat{\boldsymbol{\mu}}_{\boldsymbol{r},t}^q, \hat{\boldsymbol{\sigma}}_{\boldsymbol{r},t}^q] = \psi^{q,\boldsymbol{r}}(d_{\boldsymbol{r},t}) \text{ where } d_{\boldsymbol{r},t} = [pl_t, \psi^{\boldsymbol{r},t}(\boldsymbol{r}_t)] \tag{5}$$

The higher layers take information from lower layers and compute:

$$[\hat{\boldsymbol{\mu}}_{c,t}^q, \hat{\boldsymbol{\sigma}}_{c,t}^q] = \psi^{q,c}(d_{c,t}) \text{ where } d_{c,t} = [d_{(-),t}, \psi^{c,t}(c_t)] \tag{6}$$

Here we have $c \in \{\mathbf{a}, \mathbf{s}\}$, and $d_{(-),t}$ denotes deterministic outputs from a lower layer. Similar to the generator, neural functions $\psi^{(\cdot)}$ are implemented by MLP with batch normalization.

We then implement a stochastic downward pass to recursively compute the approximate posterior. At the top layers, the Gaussian posterior applies the estimated parameters from a deterministic function:

$$q(\mathbf{z}_{\mathbf{s},t}|\mathbf{s}_t, pl_t) = \mathcal{N}(\boldsymbol{\mu}_{\mathbf{s},t}^q, \boldsymbol{\sigma}_{\mathbf{s},t}^q) \text{ where } [\boldsymbol{\mu}_{\mathbf{s},t}^q, \boldsymbol{\sigma}_{\mathbf{s},t}^q] = [\hat{\boldsymbol{\mu}}_{\mathbf{s},t}^q, \hat{\boldsymbol{\sigma}}_{\mathbf{s},t}^q] \tag{7}$$

At the lower layers, the inference model applies a precision-weighted combination of $(\hat{\boldsymbol{\mu}}_{c,t}^q, \hat{\boldsymbol{\sigma}}_{c,t}^q)$ carrying bottom-up information and $(\boldsymbol{\mu}_{c,t}^p, \boldsymbol{\sigma}_{c,t}^p)$ from the generative distribution carrying top-down prior information. The approximate posteriors are computed by:

$$q(\mathbf{z}_{c,t}|c_t, \mathbf{z}_{(+),t}, pl_t) = \mathcal{N}(\boldsymbol{\mu}_{c,t}^q, \boldsymbol{\sigma}_{c,t}^q) \text{ where} \tag{8}$$
$$\boldsymbol{\mu}_{c,t}^q = \frac{\hat{\boldsymbol{\mu}}_{c,t}^q(\hat{\boldsymbol{\sigma}}_{c,t}^q)^{-2} + \boldsymbol{\mu}_{c,t}^p(\boldsymbol{\sigma}_{c,t}^p)^{-2}}{(\hat{\boldsymbol{\sigma}}_{c,t}^q)^{-2} + (\boldsymbol{\sigma}_{c,t}^p)^{-2}} \text{ and } \boldsymbol{\sigma}_{c,t}^q = \frac{1}{(\hat{\boldsymbol{\sigma}}_{c,t}^q)^{-2} + (\boldsymbol{\sigma}_{c,t}^p)^{-2}}$$

Here we have $c \in \{\mathbf{a}, \boldsymbol{r}\}$, and $\mathbf{z}_{(+),t}$ denotes latent variables from the upper layer. This parameterization has a probabilistic motivation by viewing $\hat{\boldsymbol{\mu}}_{c,t}^q$ and $\hat{\boldsymbol{\sigma}}_{c,t}^q$ as the approximate Gaussian likelihood that is combined with a Gaussian prior $\boldsymbol{\mu}_{c,t}^p$ and $\boldsymbol{\sigma}_{c,t}^p$ from the generative distribution. Together these form the approximate posterior distribution $q(\mathbf{z}|\mathbf{z}_{(+)}, c)$ using the same top-down dependency structure for both inference and generation.

Based on [36], the timestep-wise variational lower bound for our model is :

$$\sum_{t=1}^{T}\Big\{\sum_{c \in \{\mathbf{s}, \mathbf{a}, \boldsymbol{r}\}}\Big[-\beta\mathcal{D}_{\mathrm{KL}}[q(\mathbf{z}_{c,t}|c_t, \mathbf{z}_{(+),t}, pl_t)\|p(\mathbf{z}_{c,t}|c_t, \mathbf{z}_{(+),t})]\Big] + \mathbb{E}_{\mathbf{z}_{\boldsymbol{r},t}\sim q(\mathbf{z}_{\boldsymbol{r},t}|\cdot)}\Big[\log p(pl_t|\mathbf{z}_{\boldsymbol{r},t}) - \\ \lambda^\zeta\mathcal{L}^\zeta(\mathbf{z}_{\boldsymbol{r},t}, \mathbf{s}_t, \mathbf{a}_t, \boldsymbol{r}_t)\Big]\Big\} \tag{9}$$

where $\beta$ controls the scale of KLD regularization. To mitigate local optima caused by posterior collapse ($\mathcal{D}_{\mathrm{KL}}(\cdot)$ drops to 0) at the initial stage of training [18], we apply a warm-up from deterministic to variational encoder by scaling $\beta$ from 0 to 1 [16]. The bottom layer latent variables $\mathbf{z}_{\boldsymbol{r},t}$ absorb context and player information from upper layer and form a contextualized player representation:

$$q(\mathbf{z}_t|\mathbf{s}_t, \mathbf{a}_t, \boldsymbol{r}_t, pl_t) = q(\mathbf{z}_{\boldsymbol{r},t}|\cdot).$$

This real-valued vector can replace the one-hot player representation and facilitates downstream applications such as predicting expected goals or score differences (see Section 5.3). We also add an application loss $\mathcal{L}^\zeta$ with a parameter $\lambda^\zeta$ to control its scale. This loss combines the gradient of the application models with the embedding inference. Co-training the embedding model and the application model significantly accelerates training and dynamically incorporates player information into different downstream applications.

# 5 Empirical Evaluation

We evaluate the generative performance of the embedding models for player identification and study the usefulness of embeddings for application tasks [40, 41]. Our application tasks include 1) estimating expected goals and 2) predicting the final score differences, which are among the most challenging tasks in sports analytics [42, 8].

## 5.1 Experiment Settings

We introduce our ice-hockey dataset and comparison methods following an ablation design. The Appendix gives further details about experimental settings and implementations.

**Dataset:** We utilize a dataset constructed by Sportlogiq. The data provides information about **game events** and **player actions** for the entire 2018-2019 National Hockey League (NHL) season, which contains 4,534,017 events, covering 31 teams, 1,196 games and 1,003 players. The dataset consists of events around the puck. Each event includes the identity and action of the player possessing the puck, with time stamps and features of the game context. (We provide a complete list of game features in Appendix.) The dataset records which unique player possesses the puck. In this paper, we refer to the acting player as the on-puck player. We randomly divide the dataset containing 1,196 games into a training set (80%), a validation set (10%), and a testing set (10%) and implement 5 independent runs. The resulting means and variances are reported.

**Comparison models:** We employ an ablation design that removes different components from our full VaRLAE system. We first remove the hierarchical dependency of latent variables and train a Conditional Variational Recurrent Neural Network (**CVRNN**) [36]. CVRNN concatenates the context variables ($\mathbf{s}_t$, $\mathbf{a}_t$ and $\mathbf{r}_t$) and applies a single layer of latent variables to embed players with variational inference. We then replace the variational encoder with a Conditional Auto-Encoder at each RNN cell (**CAERNN**) that learns a deterministic player representation. The third model is a Conditional Variational Auto-Encoder (**CVAE**) [33] that discards the play history and conditions only on the current observations with no recurrence. Replacing the variational model in CVAE with a Deterministic Encoder yields (**DE**) player embedding [8]. DE is a regressor that directly maps the current observations to the acting player. We also compare our player representation framework to traditional policy embedding. The implementation follows a state-of-the-art Multi-Agent Behavior Encoder (**MA-BE**) [9]. We present a more detailed summary of the comparison models in the Appendix. In application tasks (Section 5.3), we also compare the options of 1) applying one-hot player identities (**Pids**) directly to 2) adding no player information (**N/A**) to application models.

## 5.2 Generative Performance of Embedding Models: On-Puck Player Identification

This experiment studies the generative performance of embedding models: predict which player is acting given the current match context. We compare our VaRLAE model to 6 baselines: 1) identifying player with embedding models: DE, CVAE, MA-BE, CAERNN and CVRNN 2) applying a RNN to model the game history and predict the acting player without a player embedding. The large player space (over 1k players) undermines the performance of encoders that do not utilize the recent play history. To make a fair comparison, our *constrained setting* limits the predictions to recently acting players: the current on-puck player (the correct answer) and the players that have possessed the puck in the previous 10 steps during testing. (10 is the trace length of our RNNs.) To study the identification performance for the players with *sparse participation*, we select the players (a total of 51 players) with fewer than 100 observations (out of a total of 4M events) and report the results.

Table 1: Results for acting players identification. We report both Accuracy and Log-Likelihood (LL).

| Prediction Method | Standard | | Constraining | | Sparse participation | |
|---|---|---|---|---|---|---|
| | Accuracy | LL | Accuracy | LL | Accuracy | LL |
| DE | 9.40% ± 3.06E-5 | -17.42 ± 2.23E-1 | 26.14% ± 6.45E-5 | -1.60 ± 1.10E-5 | 2.33% ± 6.95E-3 | -22.90 ± 0.02 |
| CVAE | 11.94% ± 2.80E-5 | -4.90 ± 2.84E-5 | 28.33% ± 2.96E-4 | -1.63 ± 7.77E-7 | 4.87% ± 8.93E-3 | -5.02 ± 0.01 |
| MA-BE | 19.74% ± 2.47E-4 | -3.08 ± 1.75E-3 | 51.75% ± 7.58E-5 | -1.59 ± 7.92E-6 | 5.11% ± 2.92E-4 | -7.61 ± 1.33 |
| RNN | 36.49% ± 2.21E-6 | -3.10 ± 2.80E-4 | 54.21% ± 2.80E-6 | -1.55 ± 2.43E-4 | 6.67% ± 5.03E-4 | -6.85 ± 2.15 |
| CAERNN | 43.64% ± 1.27E-5 | -2.11 ± 1.55E-3 | 67.43% ± 5.21E-6 | -1.38 ± 7.64E-4 | 11.65% ± 3.06E-3 | -3.96 ± 1.20 |
| CVRNN | 46.61% ± 9.08E-5 | -2.12 ± 2.27E-3 | 71.76% ± 4.02E-6 | **-1.33** ± 2.35E-6 | 24.30% ± 1.92E-3 | -9.67 ± 2.36 |
| VaRLAE | **50.01**% ± 2.56E-6 | **-1.76** ± 1.29E-3 | **78.54**% ± 3.62E-6 | **-1.33** ± 5.16E-4 | **36.65**% ± 2.13E-4 | **-2.99** ± 0.63 |

Table 1 shows the results for acting player identification. VaRLAE achieves leading performance on both the prediction accuracy and the log-likelihood. *The improvement is most apparent ($> 10\%$) for the players with sparse participation*, which demonstrates VaRLAE is more robust to unbalanced player participation. For the variational encoders, they perform better than the deterministic encoders. This is because the shrinkage regularizer prevents overfitting to the distribution of popular players in the training dataset. Game history is another crucial aspect for player identification, allowing the RNN models to outperform the memoryless models with observations only from the current time step. Constraining the candidate players to the group of recent on-puck players significantly improves the identification performance. The difference is most apparent for MA-BE (improves $> 30\%$), which indicates that policy embeddings do not distinguish individual players sufficiently.

**Embedding Visualization and Case Study:** We visualize the generated player embeddings as follows. 1) Randomly select 5 games from the testing set. 2) Compute the contextualized embeddings for the acting player at each event. 3) Visualize the high-dimensional embedding with the unsupervised T-distributed Stochastic Neighbor Embedding (T-SNE) [43]. Figure 2 illustrates the scatter plots. The embeddings computed by our VaRLAE (top plots) show a *shrinkage effect*.

*Player Positions.* VaRLAE embeddings are similar for players in the same position (left column), as they are more likely to perform similarly. Although the position information is masked during training, our model manages to infer it from players' behavior and assigns closer embeddings to players in the same position. For a case study on defense men (middle column), we select 5 players who are well-known and perform a similar number of actions in our 5 selected games. The plot shows that while defencemen tend to be mapped to similar embeddings, our encoder also learns which contexts distinguish them from each other.

*Action Locations.* Action locations are part of the state variable $\mathbf{s}_t$ conditioning player embeddings (right column). VaRLAE embeddings are similar for players when they act in the same zone. The similarity is weaker for the CAERNN embeddings (bottom plots). Without shrinkage, CAERNN overfits; it over-emphasizes outliers in the training data, which prevents generalizing to player behavior in the test data. The Appendix illustrates the influence of *action types* on embeddings.

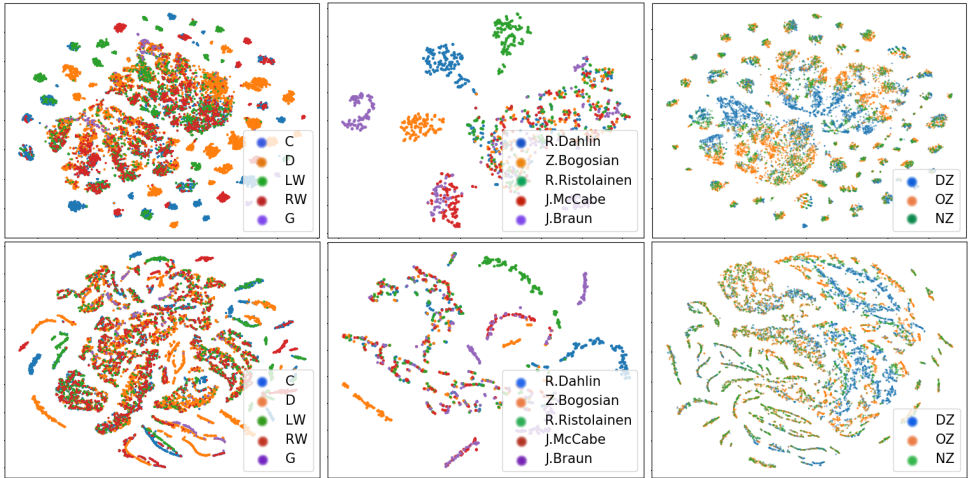

Figure 2: Embedding visualization. Each data point corresponds to a player embedding conditioning on the game context at the current time step $t$. The player embeddings are labelled 1) player positions (on the *left* column, including Center (C), Defense (D), Left-Wing (LW) and Right-Wing (RW) and Goalie (G)) 2) 5 selected defence men (on the *middle* column) and 3) player locations (on the *right* column, including Defence Zone (DZ), Neutral Zone (NZ) and Offensive Zone (OZ)). The embeddings are computed by VaRLAE (*top plots*) and CAERNN (*bottom plots*) respectively.

## 5.3 Predictive Performance On Application Tasks

We validate our VaRLAE model by feeding the generated embeddings to other application models and studying their influence on the model performance in practical application tasks.

**Expected Goals Estimation:** We validate the player embeddings for the practical task of estimating the Expected Goal (XG). An XG metric is a shot quality model that weights each *shot* by its chance of leading to a goal [42]. We incorporate $\mathbf{z}_t$ from the learned player representations into XG prediction. At time $t$, we input $\mathbf{s}_t, shot_t, \mathbf{z}_t$ to the application model $\zeta$ (implemented as a RNN), which is trained to output $p(goal_{t+1}|\mathbf{s}_t, shot_t, \mathbf{z}_t)$: the probability that a goal is scored after the player $pl_t$ makes the shot. Our dataset provides ground-truth labels for whether a given shot led to a goal. Since only a few shot attempts lead to a goal ($<3.9\%$), the training data is highly imbalanced. We employ a resampling method [44] so that successful and failed shots occur equally often during training.

Table 2: Expected goal results applying different player embeddings.

| Player Embedding Method | Performance | | | |
|---|---|---|---|---|
| | Precision | Recall | F1-score | AUC |
| N/A | $0.12 \pm 1.75$E-4 | $0.79 \pm 9.46$E-4 | $0.21 \pm 4.26$E-4 | $0.86 \pm 3.56$E-4 |
| Pids | $0.09 \pm 1.62$E-4 | $0.62 \pm 2.52$E-3 | $0.15 \pm 4.20$E-4 | $0.70 \pm 1.25$E-3 |
| DE | $0.30 \pm 1.26$E-4 | $0.92 \pm 4.21$E-4 | $0.45 \pm 1.87$E-4 | $0.96 \pm 1.86$E-5 |
| CVAE | $0.33 \pm 5.33$E-5 | $0.95 \pm 8.72$E-5 | $0.49 \pm 8.29$E-5 | $0.96 \pm 1.13$E-6 |
| MA-BE | $0.35 \pm 1.44$E-4 | $0.91 \pm 2.30$E-4 | $0.50 \pm 1.56$E-4 | $\mathbf{0.97} \pm 1.46$E-6 |
| CAERNN | $0.29 \pm 1.05$E-4 | $0.96 \pm 1.56$E-4 | $0.44 \pm 1.65$E-4 | $0.95 \pm 1.27$E-5 |
| CVRNN | $\mathbf{0.40} \pm 4.75$E-4 | $0.84 \pm 1.81$E-4 | $0.54 \pm 2.97$E-4 | $0.96 \pm 4.07$E-6 |
| VaRLAE | $0.37 \pm 2.01$E-4 | $\mathbf{0.98} \pm 1.32$E-4 | $\mathbf{0.54} \pm 8.14$E-5 | $0.96 \pm 2.23$E-6 |

Using the most likely label as the predicted class, Table 2 shows the accuracy results on the testing set. Without including any player information (N/A), predictions have very limited precision, because the model does not have access to player information that would allow it to distinguish above-average from below-average shooters. Adding the pids to the input does provide this information, but the prediction model fails to take advantage of it. This shows that player information is difficult to utilize from a sparse one-hot representation. Applying the embeddings from a player encoder (e.g. DE and CVAE) improves both precision and recall, but the improvement is limited by the lack of game history information. Among the recurrent models, VaRLAE achieves the highest recall and F1-score with good precision. This is because shooting strength correlates with the learned player types, which are most accurately represented by our model.

**Score Difference Prediction:** Dynamic Score Difference Prediction (DSDP) is a recently introduced task [8]: predict the final score difference $SD(T)$ under a game context $(\mathbf{s}_t, \mathbf{a}_t)$ where $t$ runs from 0 to $T$ (game ends). In preliminary experiments, we observed that traditional supervised learning methods suffer a large training variance (especially early in the game when many outcomes are equally likely). To exploit the temporal dependencies between score differences at successive times, we apply reinforcement learning; specifically the temporal difference method Sarsa prediction [45]. Sarsa learns a Q-function for a generic home/away team to estimate the expected cumulative goal scoring: $Q_{team}(\mathbf{s}_t, \mathbf{a}_t) = \mathbb{E}(\sum_{\tau=t}^{T} g_{team,\tau})$ where $team = Home/Away$ and $goal_{team}$=1 if the team scores at $t$ and 0 otherwise. From Q-functions, the Predicted Score Difference (PSD) at $t$ is given by $PSD(t) = Q_{Home}(\cdot) - Q_{Away}(\cdot) + SD(t)$. Our *application model* $\zeta$ is a DRQNN [37] that computes the Q-functions. The inputs are state $\mathbf{s}_t$, action $\mathbf{a}_t$ and the embedding $\mathbf{z}_t$ for the acting player $pl_t$. For each testing game $m$ and time $t$, the absolute error is given by $|PSD(t_m) - SD(T_m)|$. For each game time $t$, Figure 3 plots the mean and the variance of the absolute error over all testing games $m = 1, \ldots, M$. The plot shows a larger difference between real and predicted SDs at the beginning of a game, but the mean and variance of the difference become smaller towards the game end. Among the evaluated embedding methods, our *VaRLAE (the black line) manages to generate the player representations that lead to the most accurate predictions.* We also find that the accuracy advantage is strongest towards the early game, especially compared to the N/A and pids. This indicates that an informative player representation significantly alleviates the difficulty of predicting multiple outcomes early in the game. Averaging over game times $t$ defines the game prediction error for each tested game $m$. Table 3 reports the mean and standard deviation for the game prediction error. The VaRLAE embeddings yield the lowest Mean Absolute Error (MAE).

# 6 Conclusion

Capturing what team sports players have in common and how they differ is one of the main concerns of sports analytics. This work introduces a *player representation via player generation* framework that

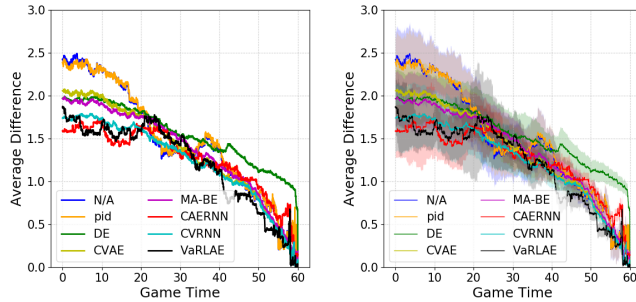

Figure 3: Temporal illustrations of the absolute error between predicted score differences and final score differences. The plots report mean (left) and mean±variance of the differences (right). Appendix shows the separated plots for each method.

Table 3: The test set game prediction error between predicted and final score differences for the entire game.

| Method | MAE |
|--------|-----|
| N/A | $1.55 \pm 0.35$ |
| Pid | $1.56 \pm 0.32$ |
| DE | $1.48 \pm 0.24$ |
| CVAE | $1.49 \pm 0.29$ |
| MA-BE | $1.45 \pm 0.25$ |
| CAERNN | $1.31 \pm 0.23$ |
| CVRNN | $1.32 \pm 0.27$ |
| VaRLAE | $\mathbf{1.28} \pm 0.29$ |

learns deep contextualized representations for ice hockey players. We described a VaRLAE model for sports data, based on a Markov Game model representation. The ELBo loss (Equation (9)) induces a shrinkage effect such that similar players are mapped to similar representations in similar match contexts. We validate the player representation on two downstream applications that are important in sports analytics: predicting expected goals and final match score differences. While our evaluation focuses on ice hockey, our approach is general and can be applied to other team sports, or indeed any strategic setting that can be modelled as a Markov game with many participants. A direction of future work is to combine player representations with modelling player interactions. Another is learning representations for different lineups, given a dataset with full observations of on-court players.

## 7 Broader Impact

We expect the main impact outside of the machine learning community to be in professional sports. As an entertainment industry, professional sports increases the quality of life for many people. With respect to the broad challenges of our society (polarization, inequality, bias), entertainment is in our view neutral. The main stakeholders impacted will be sports teams, managers, and athletes. For sports stakeholders, we expect mainly positive and some negative outcomes.

**Positive Outcomes.** Our work significantly enhances the reliability of the machine learning algorithm on modeling complex sport dynamics. This allows teams, coaches, and fans to better understand a player's style, influence, and contributions. The result will be better and even more enjoyable sports. For sports stakeholders, the main positive outcome will be that coaches can deploy players more effectively, which will help them display and improve their considerable skills. This will create winners and losers among the players, but on the whole, a A data-driven approach will increase fairness and objectivity in player performance ranking, while decreasing bias. Bias towards underrepresented groups is known to hurt the perception and career chances of underrepresented groups, such as indigenous ice hockey players; Fred Sasakamoose is a prominent example.

**Negative Outcomes.** Putting players' performance and contribution under intense study may lead to more performance pressure on the professional players. We believe that for most players this Taylorist pressure [46] is outweighed by the guidance for how to improve both their play and their market value. To apply our model, sports team should invest in technical analytic resources, which might not be affordable for small clubs. It potentially increases the inequality between top-ranked teams and lower-ranked teams. To help level the analytics playing field, we have placed our code in the public domain at github[1].

## Acknowledgments and Disclosure of Funding

This project was supported by a Strategic Project Grant from the Natural Sciences and Engineering Research Council of Canada. We are grateful for helpful comments from Jesse Davis, Tim Swartz, and the reviewers and participants in the 2019 AAAI Team Sports workshop. Our computations were facilitated by a GPU donation from NVIDIA.

## Footnotes

[1]`https://github.com/Guiliang/player-embedding-ice-hockey`

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
