[Supplementary Material · Appendix.pdf]

# Appendix

## A   Experiment Details

We provide an introduction to the dataset, the implementation details, and the comparison methods.

### A.1   Dataset Details

In this paper, we apply a play-by-play dataset constructed by NAME (name withheld to preserve anonymity). They capture the information of an on-puck player (player possessing the puck) from broadcast videos with computer version techniques. Table 4 shows a complete set of features.

Table 4: The complete list of features. The table utilizes adjusted spatial coordinates where negative numbers denote the defensive zone of the acting player and positive numbers denote his offensive zone. Adjusted X-coordinates run from -100 to +100 and Adjusted Y-coordinates from 42.5 to -42.5, where the origin is at the ice center.

| Type | Name | Range |
|---|---|---|
| Spatial Features | X Coordinate of Puck | [-100, 100] |
| | Y Coordinate of Puck | [-42.5, 42.5] |
| | Velocity of Puck | $(-\infty, +\infty)$ |
| | Angle between the puck and the goal | $[-3.14, 3.14]$ |
| Temporal Features | Game Time Left | [0, 3,600] |
| | Event Duration | $(0, +\infty)$ |
| In-Game Features | Score Differential | $(-\infty, +\infty)$ |
| | Manpower Situation | {Even Strength, Shorted Handed, Power Play} |
| | Home or Away Team | {Home, Away} |
| | Action Outcome | {successful, failure} |

### A.2   Implementation Details.

**Running settings.**   All the models (including our VaRLAE and other comparison models) are implemented by Tensorflow 1.15 (the source code has been uploaded as supplementary materials). The models are trained by mini-batch stochastic gradient descent (with batch size 32) applying Adam optimizer. The learning rate is set to 1E-5. To validate our model, we randomly divide the NHL dataset containing 1,196 games into a training set (80%), a validation set (10%), and a testing set(10%) and implement 5 independent runs. In each run, the models are trained for a total of 10 epochs (over 40M events), and testing is implemented by the hold-out validation. In this paper, we report the results in the format of mean $\pm$ variance computed across these 5 runs. All the experiments are run on a local machine with 32 GB main memory, a TITAN X GPU (12 GB memory), and a GeForce GTX 1080 GPU (12 GB memory).

18  **Model settings.**  Within our VaRLAE , the dimensions of latent variables $\mathbf{z_s}$, $\mathbf{z_a}$ and $\mathbf{z_r}$ are set to
19  64, 64 and 32 respectively (following [1]). The sizes of other hidden layers (in both LSTM and MLP)
20  are set to 256. The max trace length of LSTM is set to 10 following previous works [2, 3].

## A.3  A Summary for Comparison methods

22  Table 5 summarizes the differences between these comparison methods and our model. our VaR-
23  LAE applies a hierarchy of latent variables to embed players. To make fair comparisons, we set the
24  dimension of the embedding vector to 256 for all comparison methods.

25  **N/A** indicates no player information is applied and **Pids** indicates that we directly input one-hot
26  player ids to the downstream application models.

27  **Deterministic Encoder (DE) [4]:** DE applies a conditional auto-encoder structure. DE maps the
28  current observations $(\boldsymbol{o}_t, \mathbf{a}_t, \boldsymbol{r}_t, pl_t)$ to the acting players by minimizing a reconstruction loss. It does
29  not model the game history.

30  **Conditional Variational Auto-Encoder (CVAE) [5]:** Our implementation of CVAE follows our
31  player representation framework (Section 3.2). It learns a player representation conditioning current
32  observations: $q(\mathbf{z}_t|\boldsymbol{o}_t, \mathbf{a}_t, \boldsymbol{r}_t, pl_t)$ (without modeling the game history).

33  **Multi-Agent Behavior Encoder (MA-BE)[6]:** MA-BE applies a policy embedding framework and
34  models the behavior of players by imitation learning. To identify the embedding for different agents,
35  MA-BE introduces an exponential triple loss to punish the similarity among embeddings for different
36  players. The scale of triple loss is controlled by a hyper-parameter ($\lambda$). A large $\lambda$ produces well-
37  distinguished player embeddings, but it also generates huge loss variance which leads to large gradient
38  and undermines the model convergence. When we apply MA-BE to learn the player representations,
39  we find it hard to determine such a $\lambda$ that can adequately facilitate both the player identification and
40  model convergence, given the large number of players and the unbalanced representation. In the
41  experiment, we examine different $\lambda$ and obtain a reasonable model performance when $\lambda = 0.0001$.

42  **Conditional Variational RNN (CVRNN):** CVRNN implements a VRNN [7] conditioning on the
43  game context. CVRNN includes a CVAE at each RNN cell. It models the game history with RNN
44  hidden states and learns a contextualized player representation $q(\mathbf{z}_t|\mathbf{s}_t, \mathbf{a}_t, \boldsymbol{r}_t, pl_t)$ following our
45  player representation framework (Section 3.2).

46  **Conditional Auto-Encoder RNN (CAERNN):** CAERNN applies a similar implementation to
47  CVRNN except, at each RNN cell, it replaces the Variational Auto-Encoder with a determinis-
48  tic Auto-Encoder.

Table 5: A summary of comparison methods.

|  | Hierarchical Embedding | Game History | Stochastic-Model | Continuous-Value Embedding | Player-Information | Policy-Representation |
|---|---|---|---|---|---|---|
| N/A | No | No | No | No | No | No |
| Pids | No | No | No | No | Yes | No |
| DE | No | No | No | Yes | Yes | No |
| CVAE | No | No | Yes | Yes | Yes | No |
| MA-BE | No | Yes | No | Yes | Yes | Yes |
| CAERNN | No | Yes | No | Yes | Yes | No |
| CVRNN | No | Yes | Yes | Yes | Yes | No |
| VaRLAE | Yes | Yes | Yes | Yes | Yes | No |

# B  A Spatial Illustration for the Shot Attempts

50  We randomly sample 20 games from the training data and show a spatial illustration of shots that
51  happened during these games in Figure 4. This plot is consistent with our description (section 5.3)
52  that the training data is highly imbalanced and only a few shot attempts lead to a goal. The plot also
53  shows that the locations of the successful and the unsuccessful shots are highly overlapped. Without
54  knowing the identity of the acting player, it is hard to determine whether the shot can be made or not.

Figure 4: The spatial illustration of shot attempts on a hockey rink. We apply the adjusted coordinate (see Table 4) and the play always flows from left to right. Blue circles represent unsuccessful shots and red stars indicate successful shot.

# C Additional Results

## C.1 Embedding Visualization

We visualize the embeddings generated by our VaRLAE and CAERNN to compare the difference between a stochastic encoder and a deterministic encoder. In our paper, we label the embedding with players' positions, names, and locations. Here, we complement these visualizations by further labeling the action types of players (the visualization method is explained in our paper.).

Figure 5: Embedding visualization. Each data point corresponds to a player embedding conditioning on the game context at the current time step $t$. The player embeddings are labelled by the action types. The embeddings are computed by VaRLAE (*top plots*) and CAERNN (*bottom plots*).

## C.2 Temporal Illustrations of the Absolute Error

To show more details of the predicted score difference results , we separately illustrate the mean±variance plot (Figure 3 in Section 5.3) for all the evaluated embedding methods.

| (a) N/A | (b) Pids | (c) DE | (d) CVAE |
|---|---|---|---|

| (e) MA-BE | (f) CVRNN | (g) CAERNN | (h) VaRLAE |
|---|---|---|---|

Figure 6: Temporal illustrations of the absolute error between predicted score differences and final score differences. We report mean±variance of the error at each time step for all compared methods.

## C.3 Posterior Collapse

Figure 7 shows the Kullback–Leibler Divergence (KLD) between the posterior and the context-specific prior (for the variational encoders) during training. Among the studied methods, CVAE quickly reduces KLD to a small value (around 0.0005) after training on only a few games, but its performance is less unstable without modeling the game history. CVRNN converges slower and the KLD gradually drops to a very small number (around 3E-05) after training, which indicates the prior can replace the posterior and the decoder can generate the distribution of acting player without the player representation. It is consistent without intuition that a high capacity decoder like RNN can lead to posterior collapse [8]. Our VaRLAE significantly alleviates this problem by applying a hierarchy of latent variables and a deterministic warm-up during training (Section 4). The KLD reduces smoothly until it converges a value around 0.03.

Figure 7: The KLD between the posteriors and the priors during training for VaRLAE , CVRNN and CVAE (from left to right).