[Reviews · NeurIPS 2020]

Review 1

Summary and Contributions: This paper introduces VaRLAE, a variational sequential generative model with a ladder structure for its latent variables, to learn a representation for player identity conditioned on the game state history. They evaluate the quality of learned representations on downstream tasks in ice hockey, such as predicting expected goals and the score differential. --- Post-rebuttal update --- A strength of this paper that I had previously discounted was the novelty of the dataset itself. The dataset does seem to be quite challenging, and I believe the NeurIPS community would be excited about this new domain. For this reason, I've increased my score. That said, I find the empirical evaluation of VaRLAE not quite convincingly enough. More analysis of the ladder structure (e.g. comparison with fully-contextualized ladder structure) and/or more datasets/domains would make this paper much stronger.

Strengths: 1) The problem setting is challenging and also has applications beyond just sports. For example in autonomous driving, one may wish to learn good agent embeddings conditioned on the social context. The greatly improved performance in the “sparse participation” experiment is a promising result for many domains (i.e. being able to detect when agents are behaving abnormally). I think the paper would benefit from some of this discussion. 2) Paper is well-motivated, well-written, and easy to follow.

Weaknesses: 1) The technical contribution is incremental, as it seems like the model described is a fairly straightforward combination of VRNNs with ladder network structure for latent variables. The hierarchical latent variables are what distinguishes VaRLAE from other baselines presented in the paper, so I would have liked to see some analysis of all the latent variables, not just ones at the lowest level. 2) The experimental results on the downstream tasks show marginal improvements over the best baseline. The performance using VaRLAE player representations is on par with CVRNN player representations for both expected goal estimation (comparable F2-score and AUC) and score difference prediction (lower mean, but larger variance within range). The effectiveness of the learned representations is unclear some more experiments (or domains). 3) The main takeaway for the embedding visualization in Figure 2 is also unclear. I can see that the embeddings from VaRLAE are “shrunk” together due to optimizing the KL-divergence towards a Guassian prior. On the other hand, the CAERNN embeddings don’t look as clean, but the clusters are still distinguishable from each other and they don’t perform much worse in downstream tasks (the biggest difference is F1-score for expected goal estimation). Figure 2 looks like it’s mainly highlighting the difference between variational and non-variational models. How do the embeddings compare with those from CVRNN, the best baseline? I suspect they might look similar to VaRLAE.

Correctness: Yes

Clarity: Yes

Relation to Prior Work: The related work would benefit from a discussion about hierarchical latent variable models for sequential data, as they have been explored for speech [1], music [2], and video [3], to name a few. [1] Fraccaro et al. Sequential Neural Models with Stochastic Layers. [2] Roberts et al. A Hierarchical Latent Vector Model for Learning Long-Term Structure in Music. [3] Castrejon et al. Improved Conditional VRNNs for VIdeo Prediction.

Reproducibility: Yes

Additional Feedback: How often does player identity change in a sequence? What is the frequency of the data? As a sanity check for comparison for Table 1, how would a naive baseline that merely outputs the previous on-puck player in the sequence perform? Minor comments: 1) Line 77 - “triplet” instead of “triple” 2) Line 128 - “objective” instead of “object” 3) Line 137 - does “shrinkage term” refer to the KL-divergence in the objective? 4) Line 165, Equation 4 - should the LHS of the equation on the right just be /theta_t? I assume /theta is a vector returned by /phi, and each dimension of /theta corresponds to a player. 5) Line 160 and 176 - might be more clear to have c = {s, a, r}, since both c and c_t are in the equations. 6) Line 186 - should it be q(z | z+, c)? 7) Line 275 - shot_t is repeated, one of them should be pl_t 8) Line 117, 231, 269 - extra indentation of a few spaces


Review 2

Summary and Contributions: The authors present a novel VAE architecture made to model the performance of players of Ice Hockey. Specifically, they present a Variational Recurrent Ladder Agent Encoder (VaRLAE), which biases the posterior of the learned distribution of Ice Hockey players to match certain intuitions around statistical averages and similar players. The paper introduces this architecture, includes an empirical evaluation for representing players, includes a visualization of the learned embeddings, and presents a series of experiments centered around potential applications.

Strengths: The strengths of the paper are in the technical novelty of the new VaRLAE architecture, the domain novelty of Ice Hockey, and the results across all the sets of experiments. This work is likely to be of interest to researchers interested in multi-agent modelling for these reasons. The experiments including variations on the approach is particularly helpful as a reader to gain insight into the way that each of these components acts.

Weaknesses: There are two major weaknesses with the current draft of the paper. First, the paper is very dense and at times lacking in clarity. In particular, the end of the introduction essentially walks the reader through the whole approach, but leaves out a number of important details. This was confusing to me, as a reader, as it was unclear to me when or where I might find these missing details. In addition, a large portion of the content of the back half of the introduction is repeated in the later sections, which further adds to this confusion. Second, while the authors clearly demonstrate the strength of their approach compared to its variations (which is appreciated) there are still a limited set of comparisons being made here. I would have liked to have seen comparisons to more fundamental baselines that didn't make the same assumptions, such as other recurrent models and other models meant for multi-agent modelling.

Correctness: As far as I can tell the methods and methodology are correct and sound.

Clarity: The paper is well-written at a local level. However, I found that the overall structure was somewhat confusing (as mentioned above). The introduction goes too long, bringing up topics that are under explained and introducing a lack of clarity. Section 4 is extremely dense, and could have used some of the intuitions presented in the introduction. The results of section 5, particularly at the end are comparatively rushed. I would have appreciated greater discussion of the results, and putting them into context for readers.

Relation to Prior Work: The paper does a good job of covering related prior work, and making the case for why the problem domain in question differs significantly from this prior work.

Reproducibility: Yes

Additional Feedback: I was somewhat disappointed by the broader impacts section. The authors focus almost entirely on positive outcomes. It seems to me that a model like this is likely only usable by teams with substantial technical resources or the ability to acquire those resources. As such, it may lead to an increased inequality between the top and bottom teams. In addition, given that models like this can only draw inferences from within a learned distribution there's little room for players to grow or change, meaning that a model like this may also increase inequality between players. Edit: I have read the author's response, and my review remains unchanged.


Review 3

Summary and Contributions: The authors propose a Variational Recurrent Ladder Agent Encoder (VaRLAE), a generative method for learning player representations in professional sports and evaluate their approach on professional hockey games.

Strengths: The paper is generally well written, motivated and evaluated. A nice analysis is provided in the appendix and the evaluation comparisons are fairly strong.

Weaknesses: socialGAN, SoPHie and other multi-agent representation learning approaches should be added as comparison metrics or a reason for not using them should be added as they explicitly learn individual representations with group context. Contextual information was added into these types of models in prior work (e.g. Tensor fusion) which would serve as a nice comparison for event prediction. line 129, referencing equation (9) here is a little confusing as it requires jumping to another page to understand the reference. The shot quality prediction is similar to the results reported in "“Quality vs Quantity”: Improved Shot Prediction in Soccer using Strategic Features from Spatiotemporal Data". Can the authors provide some key insights from the proposed approach that was missing in this and other prior work on shot prediction.

Correctness: Yes

Clarity: Yes

Relation to Prior Work: Comparisons to some other multi-agent representation learning approaches should be added, or a justification for not including them. This includes methods that incorporate context into the representation.

Reproducibility: Yes

Additional Feedback:


Review 4

Summary and Contributions: This paper introduces player representation to the problem of learning to predict hockey player identities from state information, predicted expected goals, and predicting final scores in hockey. The paper uses a combination of variational auto encoders, recurrence, and ladder agents (capturing markov game properties) to capture individual player dynamics for a large number of players. This may be of value in eSports analytics to infer the true world state from data and make predictions about individual players as well as entire games. The paper brings together a number of existing techniques, such as conditional variational auto-encoders, ladder agents, and recurrence. The architecture seems plausible and the results are largely in favor of the author's claims.

Strengths: theoretical grounding: the choice of existing techniques to merge together to work on the hockey dataset are well argued empirical evaluation: The paper looks at three different tasks that one might want to perform on the dataset. The dataset looks challenging with regard to the fact that there are large number of labels (1k players) that need to be differentiated to make predictions. Relevance to NeurIPS: The work may be of value in eSports application domains. The architecture presented may be of value to analogous world state tracking domains (other sports, military applications, robotics, etc) though the technique is only tested on one domain.

Weaknesses: empirical evaluation: There are number of weaknesses with regard to the empirical evaluation. Most directly, the expected goal results (fig 2) are not conclusive. It is unclear that the ladder aspect of the architecture is providing an improvement on this application task. There is reason to believe that the VaRLAE architecture is applicable to more domains than just hockey. It would strengthen the paper to see this applied to more sports datasets or even non-sports datasets that share similarities with respect to individual tracking (eSports for example?) with a large number of entities. The hockey dataset is fairly unique, however, with features (e.g. puck data), and this type of experimentation will tell readers how much the particular architecture is tuned to the particularities of this one dataset. The given evaluation does a nice job of ablation tests, which let the readers know what aspects of the system architecture are providing the increases in performance. The paper mentions other approaches and it might be useful to see a comparison to other papers. However, this reviewer acknowledges that the ablations may already give a general idea of how other papers will do on this dataset (given that other papers may not use this particular dataset). A direct comparison would be preferred under ideal circumstances, however. Relevance to NeurIPS: As above, more datasets and more domains will make it more clear to the NeurIPS community as to the flexibility and applicability of the proposed modeling technique.

Correctness: This reviewer has no concerns regarding correctness, although as noted above there are some questions pertaining to whether the proposed architecture is only applicable to hockey.

Clarity: The paper is well written. Figure 3 is very hard to read. There are too many plots crammed into a small space and overlapping lines and colors become very hard to distinguish.

Relation to Prior Work: The paper makes clear references to related techniques and makes clear theoretical arguments about what is missing in prior work.

Reproducibility: Yes

Additional Feedback: Broader impacts: Broader impacts section fails to point out the main application of this work is world state estimation with an emphasis on individual person tracking when dealing with a large number of known individuals. This reviewer is concerned that the proposed architecture may be applied toward activity recognition in the real world to identify and track individuals in noisy contexts. However, this reviewer also acknowledges that the current data set has a lot of features that might be unrealistic outside of sports. It would be nice to see a broader discussion on whether this concern is a plausible one in the views of the authors, who are much more aware of the possibilities and limitations of their work.

[Author Response · NeurIPS 2020]

We appreciate the reviewers for reading our paper and their constructive comments. This response letter is to clarify our
major claims according to the comments from reviewer 1, reviewer 2, reviewer 3 and reviewer 4. To save space, we first
answer some shared concerns from reviewers and then answer their specific questions separately.

**Shared Concerns:**
1) **Reviewer 2:***"I would have liked to have seen comparisons to more fundamental baselines that didn't make the same*
*assumptions, such as other recurrent models and other models meant for multi-agent modelling"*
**Reviewer 3:***"socialGAN, SoPHie and other multi-agent representation learning approaches should be added..."*
**Reviewer 4:***"The paper mentions other approaches and it might be useful to see a comparison to other papers..."*
Our comparison method includes MA-BE, which is a recently proposed multi-agent embedding model applied to
sequential data (Line 226 in submission). SocialGAN and other multi-agent methods are designed for trajectory data
and therefore not directly applicable to our event data. For example, SocialGAN describes a model to discriminate fake
from real trajectories. We mentioned modelling player interactions for play-by-play data as a topic for future work in
our conclusion.

2) **Reviewer 3:***"The shot quality prediction is similar to the results reported in ""Quality vs Quantity"... Can the*
*authors provide some key insights from the proposed approach that was missing in this and other prior work on ..."*
**Reviewer 4:***"It is unclear that the ladder aspect of the architecture is providing an improvement on this application."*
Prior work on ice hockey shot prediction does not take into account the identity of the shooter. Certainly not as part of a
general player representation framework. For instance, the scoring chance is higher for a top player v.s., an average
player under similar game context. Table 1 shows the benefits of modelling shooter-specific effects.

The ladder structure mitigates posterior collapse during training (Lines 148-156). We provide a detailed discussion and
results is in C.3 of our Appendix.

**Comments from Reviewer 1**
*"I would have liked to see some analysis of all the latent variables, not just ones at the lowest level."*
We visualize only $\mathbf{z}_{r,t}$ (at the lower level of ladder structure) because it conditions on $\mathbf{s}_t, \mathbf{a}_t, r_t$ and contains the most
complete information about each player. The latent variables at higher levels, for example $\mathbf{z}_{s,t}$, have no access to $r_t$ or
$\mathbf{a}_t$ (This is where our contextualized model differs from the traditional ladder structure). We have visualized the $\mathbf{z}_{s,t}$
and $\mathbf{z}_{a,t}$, but found them less informative so we did not include them. Specifically, latent values from the higher levels
distinguish players less, and show a smaller shrinkage effect: many points are smoothly distributed around the mean.
(Similar results were observed in the ladder VAE paper [16]). We can discuss the higher levels in the final version.

2)*"The main takeaway for the embedding visualization in Figure 2 is also unclear...How do the embeddings compare*
*with those from CVRNN, the best baseline? I suspect they might look similar to VaRLAE"*
Our main contribution is the idea of Player representation through Player Generation (Section 3). CVRNN and
VaRLAE are different architectures for implementing this fundamental idea. Since both methods use the same general
idea, we expected their visualization to look similar. In particular, both exhibit a shrinkage effect leading to similar
T-SNE projections. The key point of Figure 2 is to show the difference of a model without a shrinkage effect, namely
traditional auto-encoder (CAERNN).

3)*"The performance using VaRLAE player representations is on par with CVRNN player representations ... The*
*effectiveness of the learned representations is unclear some more experiments (or domains)".*
Our paper covered three popular tasks in the Ice hockey domain. CVRNN is indeed the strongest ablation method
implementing Representation-Through-Generation (Section 5.1). Our VaRLAE beats it by an average of 8% (over 12%
for players with sparse participation) in player identification. Expected goals results are mixed: CVRNN has higher
precision, VaRLAE has the second-best precision, and achieves overall best performance (Recall and F1-score).

**Comments from Reviewer 2**
1) *"The paper is very dense and at times lacking in clarity... The paper is well-written at a local level. However,...".*
Thank you for your suggestions which will help us improve clarity.

2) *"I was somewhat disappointed by the broader impacts section..."*
We will make our code available, to help level the analytics playing field. While technical skills do require resources,
professional scouts are even more expensive. Our model focuses only on a player's professional skills without
considering race, gender, or age, which encourages fairness and reduces bias. Extending the model to capture player
development over time is a great idea, thank you for the suggestion.

**Comments from Reviewer 4** (also Reviewer 1)
1) *"There is reason to believe that the VaRLAE architecture is applicable to more domains than just hockey...".*
It is true that our VaRLAE can be applied to other team sports, as we mention in our conclusion. We thought it was
important to provide a thorough in-depth evaluation of several tasks in one domain.

[Meta-Review · NeurIPS 2020]

Three reviewers participated in the discussion. R1 increased their score due to the domain novelty and the challenging dataset. In general, the reviewers acknowledged that the paper has applications beyond sports and is of interest to the NeurIPS community. I encourage the authors to follow R2's suggestions regarding clarity, as discussed in the rebuttal.